# Clinical implications of Type 2 diabetes on outcomes after cardiac transplantation

**Fouad Chouairi**[1☺]*, **Clancy W. Mullan**[2☺], **Ahmed Ahmed**[3], **Jasjit Bhinder**[4], **Avirup Guha**[5], **P. Elliott Miller**[4], **Ania M. Jastreboff**[6], **Michael Fuery**[4], **Maya Rose Chiravuri**[4], **Arnar Geirsson**[2], **Nihar R. Desai**[4], **Christopher Maulion**[4], **Sounok Sen**[4], **Tariq Ahmad**[4], **Muhammad Anwer**[2]

**1** Duke University Medical Center Department of Internal Medicine, Durham, NC, United States of America, **2** Division of Cardiac Surgery, Yale School of Medicine, New Haven, CT, United States of America, **3** Harvard Medical School, Boston, MA, United States of America, **4** Section of Cardiovascular Medicine, Yale School of Medicine, New Haven, CT, United States of America, **5** Harrington Heart and Vascular Institute, Case Western Reserve University, Cleveland, OH, United States of America, **6** Section of Endocrinology & Metabolism; Yale School of Medicine, New Haven, CT, United States of America

☺ These authors contributed equally to this work.
* fouad.chouairi@duke.edu

## Abstract

### Background

T2D is an increasingly common disease that is associated with worse outcomes in patients with heart failure. Despite this, no contemporary study has assessed its impact on heart transplantation outcomes. This paper examines the demographics and outcomes of patients with type 2 diabetes (T2D) undergoing heart transplantation.

### Methods

Using the United Network for Organ Sharing (UNOS) database, patients listed for transplant were separated into cohorts based on history of T2D. Demographics and comorbidities were compared, and cox regressions were used to examine outcomes.

### Results

Between January 1st, 2011 and June 12th, 2020, we identified 9,086 patients with T2D and 23,676 without T2D listed for transplant. The proportion of patients with T2D increased from 25.2% to 27.9% between 2011 and 2020. Patients with T2D were older, more likely to be male, less likely to be White, and more likely to pay with public insurance (p<0.001, all). After adjustment, T2D patients had a lower likelihood of transplantation (Hazard Ratio [HR]: 0.93, CI: 0.90–0.96, p<0.001) and a higher likelihood of post-transplant mortality (HR: 1.30, CI: 1.20–1.40, p<0.001). Patients with T2D were more likely to be transplanted in the new allocation system compared to the old allocation system (all, p<0.001).

### Conclusions

Over the last ten years, the proportion of heart transplant recipients with T2D has increased. These patients are more likely to be from traditionally underserved populations. Patients

**Data Availability Statement:** The data used in this study are the property of the United Network for Organ Sharing (UNOS). Researchers interested in the relevant data used in this study may request

access to the data by filling out a Data Use Agreement and applying for access with the Organ Procurement and Transplantation Network (OPTN) at https://urldefense.com/v3/__https://unos.org/__ ;!!OToaGQ!uTaJDWwEF55CP-ksNPJtVG4YGfYv2 s9oijLdNLyWPnVlwFZuPR2j4b4pjBXQQcE9orCi Zf7Abhv7UtIDPd9ojg$. The authors confirm they did not have any special access privileges that others would not have.

**Funding:** Dr. Miller reports funding through by the Yale National Clinician Scholars Program and by CTSA Grant Number TL1 TR001864 from the National Center for Advancing Translational Science (NCATS), a component of the National Institutes of Health (NIH). Its contents are solely the responsibility of the authors and do not necessarily represent the official view of NIH. The funders had no role in study design, data collection and analysis, decision to publish, or preparation of the manuscript.

**Competing interests:** We have read the journal's policy and the authors of this manuscript have the following competing interests: Dr. Desai works under contract with the Centers for Medicare and Medicaid Services to develop and maintain performance measures used for public reporting and pay for performance programs. He reports research grants and consulting for Amgen, Astra Zeneca, Boehringer Ingelheim, Cytokinetics, Medicines Company, Relypsa, Novartis, and SCPharmaceuticals. Dr. Jastreboff serves as a consultant for Novo Nordisk, Eli Lilly, Boehringer Ingelheim and receives research support from the American Diabetes Association, Eli Lilly, Novo Nordisk, and The NIH/ NIDDK. Dr. Ahmad is a consultant for Amgen, Cytokinetics, Relypsa, and Novartis. The other authors have no relationships Our data has not been previously presented. All authors have read and approved the manuscript, and reported conflicts of interest/disclosures do not alter our adherence to PLOS ONE policies on sharing data and materials.

with T2D have a lower likelihood of transplantation and a higher likelihood of post-transplant mortality. After the allocation system change, likelihood of transplantation has improved for patients with T2D.

## Introduction

Cardiac transplantation requires prudent recipient selection to ensure successful outcomes [1]. Among recipient comorbidities, type 2 diabetes mellitus (T2D) is associated with adverse short and long term outcomes with an increased risk of infection, renal failure, and allograft dysfunction [2–4]. Thus, T2D is viewed as a relative contraindication at some transplantation centers. However, the directionality and magnitude of the effects of T2D is not known in contemporary patients. Moreover, the prevalence of T2D has increased so gaining insight into its impact on transplantation could be particularly important in informing clinical decisions [2,3,5]. There has been no recent analysis of a national database to determine the demographics and outcomes of patients with T2D undergoing heart transplantation. Additionally, there has been no study examining the effects of the 2018 changes to the Organ Procurement and Transplantation Network (OPTN) allocation system for heart transplantation on patients with diabetes listed for transplant.

As a result, the aim of this paper was to use the United Network for Organ Sharing (UNOS) dataset to examine the demographics and outcomes of patients with T2D listed for heart transplantation, as well as to examine the effects of the new OPTN allocation system on access and outcomes in this patient population.

## Methods

### Data source

The UNOS registry is a prospectively maintained and retrospectively updated dataset consisting of all patients listed for and undergoing solid organ transplantation in the United States. This data is submitted at the time of listing and is updated at transplant and after transplant at 1-year intervals to account for post-operative outcomes, including patient death and graft failure, which are provided by the transplant centers and supplemented with the Social Security Administration Death Master File. The patient data and information was fully anonymized and consent for data use was obtained during data collection by the UNOS. This study was approved as exempt by the Yale Institutional Review Board. This paper is strictly compliant with the The International Society of Heart and Lung Transplantation Ethics Statement.

### Study population

We reviewed the UNOS registry for all patients listed for heart transplants between January 1, 2011 and June 12, 2020. Simultaneous heart and lung recipients and patients under age 18 were excluded. Patients with other simultaneous organ transplants were also excluded. Patients were stratified into cohorts based upon a history of T2D. Patients with type 1 diabetes and patients with less than 30 days of follow up were excluded. For secondary analysis of allocation change effects, patients listed between April 12, 2017 and June 12, 2020 were included. Those with initial listing prior to October 18, 2018 utilized the old allocation system while those listed after October 18, 2018 utilized the new allocation system. Patients with an initial listing before the allocation change and an end listing after the allocation change were also excluded. These

groups were then stratified into patients with and without T2D. The primary outcomes of interest were transplantation, waitlist mortality, and post-transplantation survival.

## Statistical analysis

Recipient and donor demographics, comorbidities, and outcomes were compared between cohorts using Chi-square tests for categorical variables and Mann Whitney U tests for continuous variables. Unadjusted and adjusted Cox regressions were used to examine odds of transplantation, waitlist death, and post-transplant death. Patients without T2D were used as the reference for Cox regression analysis. Models were adjusted for sex, age, donor age, body mass index, race/ethnicity, insurance payor, cardiomyopathy diagnosis, extracorporeal membrane oxygenation [at listing] (ECMO), intra-aortic balloon pump [at listing] (IABP), inotropes, ventilator status, left ventricular assist device (LVAD), right ventricular assist device (RVAD), total artificial hearts (TAH), end stage renal disease (ESRD), prior cerebrovascular accidents (CVA), malignancy, implantable cardioverter defibrillator (ICD), tobacco use, and prior cardiac surgery. Secondary analysis of the new allocation system utilized the same model for adjusted analysis. Competing risks analyses were performed for waitlist survival, death, transplantation, and removal from waitlist due to recovery. Curves from each era were compared with Gray's tests of inequality. Kaplan Meier Survival Curves were made for transplantation, waitlist mortality, and post-transplantation mortality. Analysis was performed using SPSS version 26 (IBM, Armonk, NY).

## Results

### Listing characteristics of patients with Type 2 diabetes

Between January 1st, 2011 and June 12th, 2020, we identified 9,086 patients with T2D and 23,676 patients without T2D listed for heart transplantation. Overall, there was an increase in the proportion of patients listed with T2D from 25.2% in 2011 to 27.9% in 2020 (p<0.001) (Fig 1A). Patients with T2D were significantly older (p<0.001), less likely to be female (p<0.001), and more likely to be Black or Hispanic (p<0.001). Patients with T2D were more likely to pay with public insurance and less likely to have private insurance (p<0.001) (Table 1). Patients with T2D were more likely to use tobacco, more likely to have a stroke, more likely to have ESRD, more likely to have an AICD, and more likely to have had prior cardiac surgery as comorbidities (p<0.001, all) (Table 1). In terms of cardiac support, patients

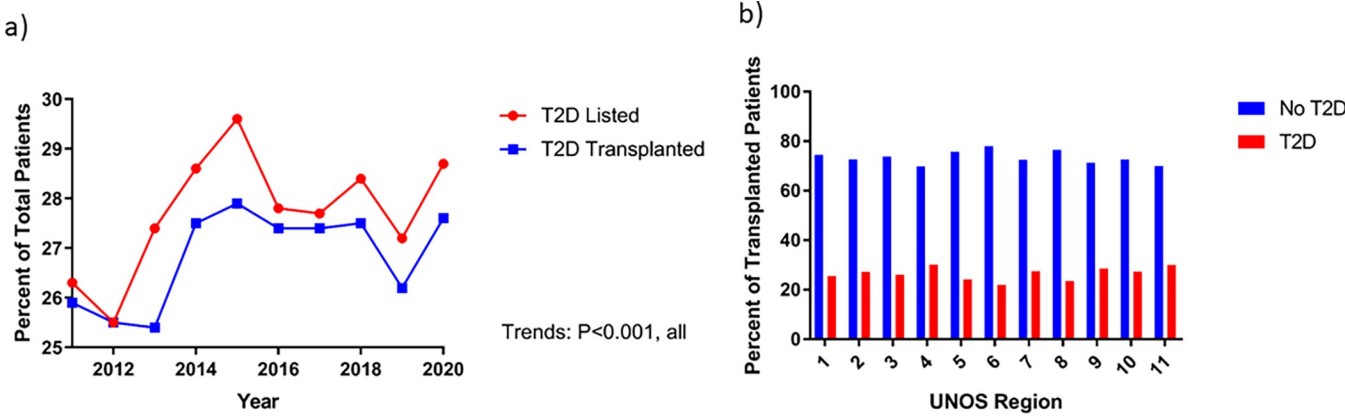

**Fig 1.** **a)** Type 2 Diabetes Prevalence Among Those Listed and Transplanted Over Time, **b)** T2D Prevalence in Transplant Recipients by Region.

**Table 1. Listing characteristics of patients according to Type 2 diabetes status.**

| Variables | No Diabetes (N = 23676) | Type 2 Diabetes (N = 9086) | P-Values |
|---|---|---|---|
| Age (yrs) | 54.0 [42.0–62.0] | 59.0 [52.0–64.0] | <**0.001** |
| Female (%) | 28.6 | 20.6 | <**0.001** |
| BMI (kg/m$^2$) | 26.8 [23.5–30.7] | 29.3 [25.8–32.7] | <**0.001** |
| **Race/Ethnicity (%)** | | | <**0.001** |
| White | 64.9 | 59.9 | |
| Black | 23.2 | 24.6 | |
| Hispanic | 7.9 | 10.0 | |
| Asian | 2.8 | 4.3 | |
| **Primary Payer (%)** | | | <**0.001** |
| Private | 52.1 | 45.2 | |
| Public | 47.1 | 54.1 | |
| **Cardiac Diagnosis (%)** | | | |
| Dilated Cardiomyopathy | 55.1 | 45.7 | <**0.001** |
| Restrictive Cardiomyopathy | 3.5 | 1.5 | <**0.001** |
| Ischemic Cardiomyopathy | 26.2 | 45.9 | <**0.001** |
| Congenital cardiomyopathy | 4.5 | 0.9 | <**0.001** |
| Hypertrophic cardiomyopathy | 3.0 | 1.0 | <**0.001** |
| Valvular Cardiomyopathy | 1.5 | 0.8 | <**0.001** |
| **Cardiac Support at Time of listing (%)** | | | |
| Inotropes | 31.0 | 31.5 | 0.404 |
| LVAD | 26.0 | 31.5 | <**0.001** |
| RVAD +/- LVAD or MCS unspecified | 1.7 | 1.3 | **0.005** |
| ECMO | 2.2 | 1.3 | <**0.001** |
| IABP | 6.1 | 6.1 | 0.854 |
| **Comorbidities (%)** | | | |
| Tobacco user | 42.8 | 51.2 | <**0.001** |
| Prior CVA | 5.9 | 7.0 | <**0.001** |
| ESRD | 2.8 | 4.3 | <**0.001** |
| AICD | 72.6 | 79.2 | <**0.001** |
| Prior Cardiac Surgery | 38.7 | 44.9 | <**0.001** |
| **Most Recent Creatinine (Prior to Listing)** | 1.2 [0.9–1.5] | 1.3 [1.0–1.6] | <**0.001** |
| **Outcomes (%)** | | | |
| Waitlist Time (IQR) | 111.0 [30.0–360.3] | 141.0 [35.0–396.0] | **0.004** |

with T2D were more likely to have an LVAD (p<0.001), but less likely to be on ECMO (p<0.001) or a ventilator (p<0.001). Regional distribution of T2D patients listed and transplanted can be found in **Fig 1B**. Initial and final listing status can be found in **S1 Fig in S1 File**. Regarding donor characteristics, donors of transplanted patients with T2D were significantly less likely to be female and had significantly higher BMIs (p<0.001, all). Additionally, patient with T2D received hearts from significantly older patients (No T2D: 30 years vs. T2D: 31 years, p<0.001). Further details regarding donor characteristics can be found in **Table 2**.

## Waitlist and post-transplantation outcomes

Results of waitlist and post-transplantation outcomes analysis are presented in **Table 3**. After adjustment, patients with T2D had a lower likelihood of transplantation than the patients without T2D (hazard ratio [HR] 0.93, (95% confidence interval [CI]: 0.90–0.96, p<0.001).

**Table 2. Donor characteristics.**

| Variables | No Diabetes (N = 15728) | Type 2 Diabetes (N = 5788) | P-Value |
|---|---|---|---|
| Age (yrs) (median IQR) | 30.0 [23.0–40.0] | 31.0 [24.0–41.0] | **<0.001** |
| Female (%) | 30.8 | 27.2 | **<0.001** |
| BMI (kg/m$^2$) (median IQR) | 26.2 [23.1–30.3] | 26.9 [23.9–31.0] | **<0.001** |
| High Risk Donor (%) | 26.4 | 26.0 | 0.599 |
| **Race/Ethnicity (%)** | | | 0.220 |
| White | 63.6 | 65.2 | |
| Black | 16.1 | 16.0 | |
| Hispanic | 16.8 | 15.6 | |
| Asian | 1.9 | 1.6 | |
| **Substance Use (%)** | | | |
| Alcohol Use | 16.3 | 17.4 | 0.049 |
| Tobacco user | 10.0 | 10.9 | 0.085 |
| Cocaine Use | 10.7 | 11.7 | **0.043** |
| Other Drug User | 40.3 | 39.7 | 0.423 |
| **Comorbidities (%)** | | | |
| Hypertension | 32.4 | 32.5 | 0.960 |
| Malignancy | 1.3 | 1.5 | 0.216 |
| Diabetes | 3.6 | 3.7 | 0.748 |
| **Infections (%)** | | | |
| Pneumonia | 69.4 | 68.2 | 0.091 |
| UTI | 12.2 | 10.6 | **0.002** |
| HCV | 2.3 | 2.8 | 0.046 |
| CMV | 61.6 | 60.2 | 0.053 |
| **Transplant Outcomes (%)** | | | |
| Ischemic Time (IQR) | 3.2 [2.4–3.8] | 3.2 [2.4–3.8] | 0.169 |
| Distance Traveled (IQR) | 99.0 [15.0–298.0 | 103.0 [15.0–299.0] | 0.564 |
| **Donor Cause of Death (%)** | | | 0.415 |
| Anoxia | 33.4% | 33.9% | |
| Stroke | 16.9% | 17.6% | |
| Head Trauma | 47.0% | 46.0% | |
| Other/Unspecified | 0.5% | 0.5% | |

There was no significant difference in waitlist mortality when the raw, unadjusted data was analyzed (No T2D: 4.8%, T2D 5.1%, p = 0.300). After adjustment there was no significant difference in waitlist death; however, patients with T2D had a significantly higher likelihood of post-transplant death (HR: 1.30 (CI: 1.20–1.40), p<0.001). In terms of acute complications post-transplantation, patients with T2D were more likely to require dialysis (15.7% vs. 12.1%, p<0.001) or a blood transfusion (22.9% vs. 20.4%, p<0.001) (**Table 4**). Additionally, patients with T2D who were transplanted also had higher rates of graft failure post-surgery (T2D: 20.2% vs. No T2D: 15.8%, p<0.001). One- year (T2D: 10.4% vs. No T2D: 7.9%, p<0.001) two year (T2D: 13.0% vs. No T2D: 10.1%, p<0.001), and three year (T2D: 15.0% vs. No T2D 11.6%, p<0.001) post-transplant mortality were greater for patients with T2D (**S2 Fig** in S1 File). Waitlist mortality rates can be found in Table 3.

Kaplan Meier curves for transplantation, waitlist survival, and post-transplantation survival can be found in **Fig 2A and 2B**. **Fig 3** demonstrates the competing risks for waitlist outcomes among patients listed for transplant with and without T2D. T2D patients had a lower rate of

**Table 3. Cox regression for outcomes.**

| | Unadjusted Hazard Ratio | P-Value | Adjusted Hazard Ratio* | P-Value |
|---|---|---|---|---|
| Transplanted | | | | |
| No Diabetes | Reference | | Reference | |
| Type 2 Diabetes | 0.94 (0.92–0.97) | <0.001 | 0.93 (0.90–0.96) | <0.001 |
| Waitlist Death | | | | |
| No Diabetes | Reference | | Reference | |
| Type 2 Diabetes | 1.13 (1.03–1.24) | 0.009 | 1.08 (0.98–1.20) | 0.104 |
| Post-Transplant Death | | | | |
| No Diabetes | Reference | | Reference | |
| Type 2 Diabetes | 1.37 (1.28–1.47) | <0.001 | 1.30 (1.20–1.40) | <0.001 |
| **Analysis by Allocation System** | | | | |
| **Patients without diabetes** | | | | |
| Transplanted | | | | |
| Old Allocation System | Reference | | Reference | |
| New Allocation System | 1.38 (1.31–1.47) | <0.001 | 1.35 (1.27–1.43) | <0.001 |
| Post-Transplant Mortality | | | | |
| Old Allocation System | Reference | | Reference | |
| New Allocation System | 1.32 (1.07–1.62) | 0.009 | 1.21 (0.97–1.51) | 0.089 |
| **Patients with Diabetes** | | | | |
| Transplanted | | | | |
| Old Allocation System | Reference | | Reference | |
| New Allocation System | 1.33 (1.21–1.46) | <0.001 | 1.33 (1.20–1.47) | <0.001 |
| Post-Transplant Mortality | | | | |
| Old Allocation System | Reference | | Reference | |
| New Allocation System | 1.47 (1.10–1.95) | 0.009 | 1.62 (1.19–2.19) | 0.002 |
| **New Allocation System Only** | | | | |
| Transplanted | | | | |
| Without Diabetes | Reference | | Reference | |
| With Diabetes | 0.94 (0.87–1.01) | 0.105 | 1.00 (0.92–1.08) | 0.920 |
| Post-Transplant Mortality | | | | |
| Without Diabetes | Reference | | Reference | |
| With Diabetes | 1.52 (1.18–1.96) | 0.001 | 1.37 (1.04–1.79) | 0.024 |

*adjusts for sex, age, donor age, BMI, race/ethnicity, insurance payor, cardiomyopathy diagnosis, ECMO, IABP, inotropes, ventilator status, LVAD, RVAD, TAH, ESRD, CVA, malignancy, AICD, tobacco use, and prior cardiac surgery.

**Table 4. Acute complications post-transplantation.**

| Variables | No Diabetes (N = 15728) | Type 2 Diabetes (N = 5788) | P-Value |
|---|---|---|---|
| Airway Dehiscence | 0.3 | 0.3 | 0.615 |
| Stroke | 2.9 | 3.3 | 0.151 |
| Dialysis | 12.1 | 15.7 | <0.001 |
| Pacemaker | 2.7 | 2.7 | 0.837 |
| Steroids | 7.2 | 6.9 | 0.444 |
| Transfusion | 20.4 | 22.9 | <0.001 |

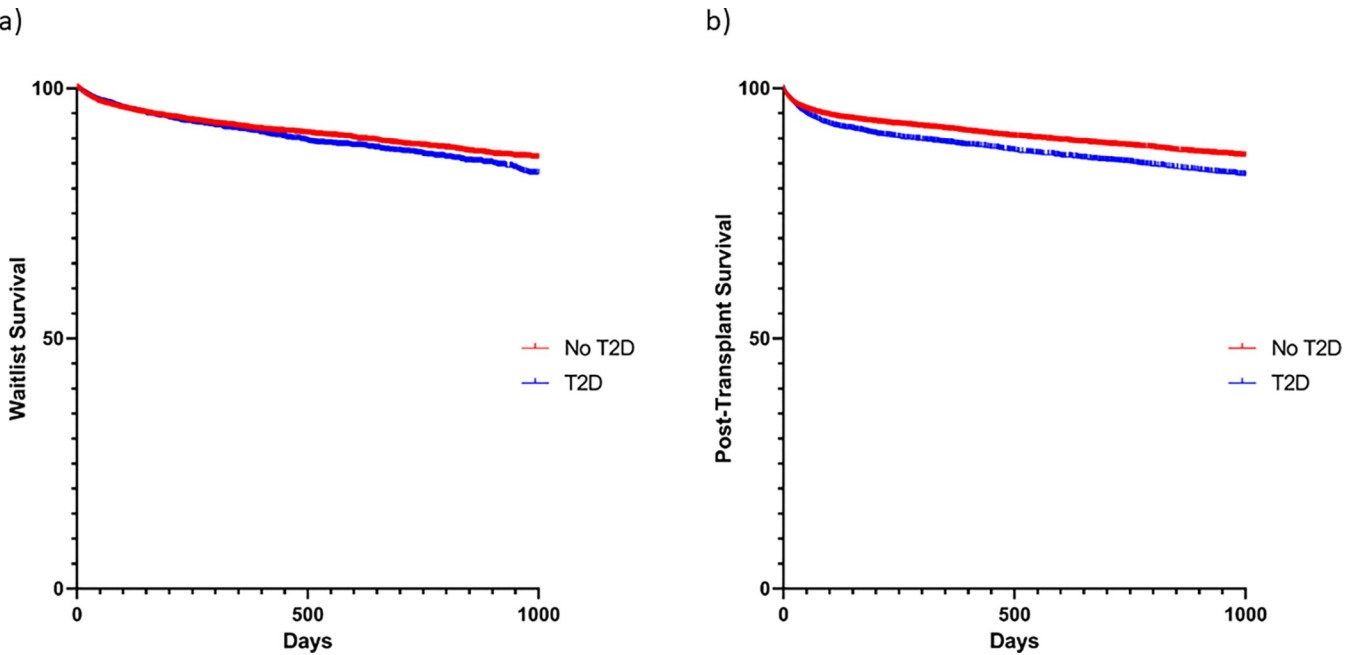

**Fig 2.** Kaplan meier analysis for waitlist survival, and post-transplantation survival **(a)** Waitlist Survival **(b)** Post-Transplant Survival.

transplantation (p<0.001), greater waitlist mortality (p<0.001), and were less likely to be removed from the waitlist for recovery (p = 0.006). Most common causes of post-transplant mortality can be found in **Table 5**.

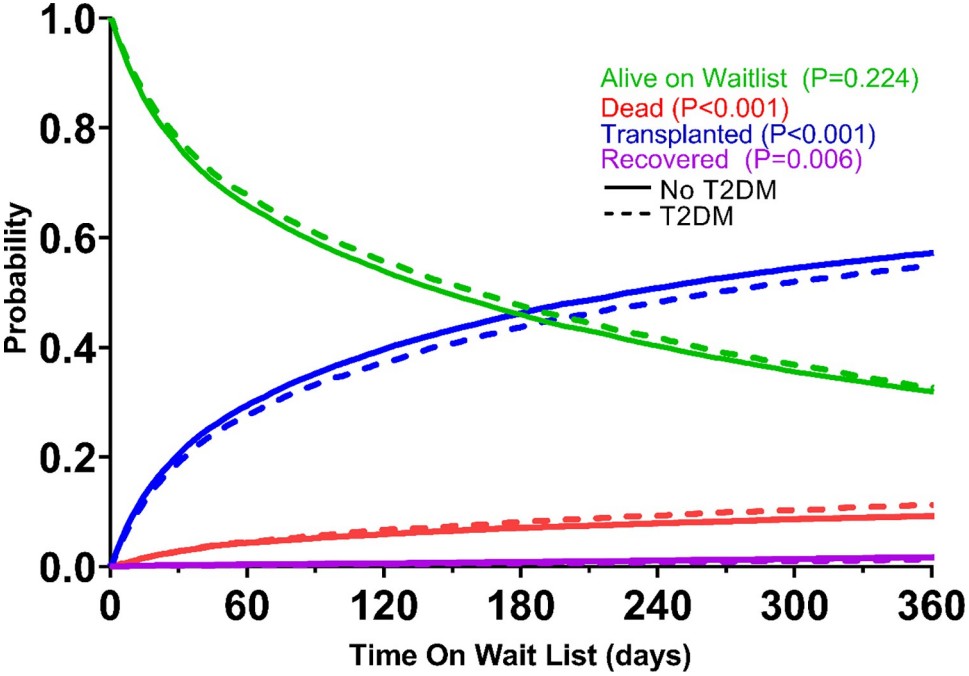

**Fig 3. Competing risks in patients with Type 2 diabetes undergoing transplantation.**

## Allocation system analysis

Among patients with and without T2D, the median age of listed patients decreased with the new allocation system (p<0.001). Among patients with T2D, the new allocation system saw more Black and Hispanic patients listed (p<0.001) and fewer patients with ischemic cardiomyopathy listed (p = 0.028). There was a decreased utilization of LVADs in the new allocation system (p<0.001), but an increased utilization of both IABP (p<0.001) and ECMO (p = 0.005). With the new allocation system, waitlist times had significantly decreased (47 days vs. 23 days, p<0.001) while ischemic time (3.1 hours to 3.5 hours, p<0.001) and travel distance (85 miles vs. 238 miles, p<0.001) had both increased significantly among patients with T2D (**Table 6**).

With the new allocation system, there was an increased adjusted odds of transplantation among both patients with T2D (HR: 1.22, 96% CI: 1.20–1.47, p<0.001) and patients without T2D (HR: 1.35, 95% CI: 1.27–1.43, p<0.001). However, there was increased adjusted post-transplant mortality for patients with T2D in the new allocation system (HR: 1.562, 95% CI: 1.19–2.19, p = 0.002) and in the new allocation system alone, T2D remained a risk for post-transplant mortality (HR: 1.37, (CI: 1.04–1.79), p = 0.024). Interestingly, when isolated to just the new allocation system, there was no significant difference in the likelihood of transplantation based on T2D status (**Table 3**).

## Discussion

This study examined the impact of T2D among contemporary patients listed for heart transplantation and we found that patients with T2D were less likely to receive a transplant and had

**Table 5. Causes of mortality.**

| Variables | No Diabetes (N = 2308) | Type 2 Diabetes (N = 1118) |
|---|---|---|
| **Graft Failure** | **9.2** | **7.6** |
| Graft Failure: Primary Failure | 5.9 | 4.9 |
| Graft Failure: Acute Rejection | 1.9 | 1.9 |
| Graft Failure: Non-Specific | 1.4 | 0.8 |
| **Infection** | **12.7** | **15.6** |
| Infection: Bacterial Septicemia | 6.6 | 9.7 |
| Infection: Bacterial Pneumonia | 1.5 | 1.5 |
| Infection: Fungal- Aspergillus | 1.4 | 0.7 |
| Infection: Fungal- Other | 1.7 | 2.0 |
| Infection: Other | 1.5 | 1.7 |
| Cardiac Arrest | 10.7 | 7.3 |
| Cardiogenic Shock | 2.7 | 2.6 |
| Cardiovascular Death | 2.0 | 2.0 |
| Respiratory Failure | 3.8 | 4.6 |
| Stroke | 2.6 | 2.6 |
| Intracerebral Hemorrhage | 1.7 | 0.9 |
| Brain Anoxia | 2.0 | 2.6 |
| Metastatic Malignancy | 3.5 | 2.6 |
| Primary Malignancy | 1.0 | 1.5 |
| Malignancy Other | 1.3 | 2.3 |
| Renal Failure | 1.0 | 0.9 |
| Multiple Organ Failure | 11.2 | 10.0 |

*Only included cause with an incidence greater than or equal to 1.0%.

**Table 6. Differences based on allocation systems.**

| Variables | No Diabetes | | | | Type 2 Diabetes | | |
| --- | --- | --- | --- | --- | --- | --- | --- |
| | Pre-Allocation System Changes (N = 2688) (Apr 1, 2017-Oct 12, 2018) | Post-Allocation System Changes (N = 2925) (Oct 12, 2018-Jun 12, 2020) | P-Value | | Pre-Allocation System Changes (N = 1055) (Apr 1, 2017-Oct 12, 2018) | Post-Allocation System Changes (N = 1072) (Oct 12, 2018-Jun 12, 2020) | P-Value |
| Age (yrs) | 56.0 [44.0–63.0] | 54.0 [42.0–63.0] | **0.006** | | 60.0 [53.0–65.0] | 59.0 [52.0–64.0] | **0.018** |
| Female (%) | 30.4 | 31.0 | 0.618 | | 24.1 | 23.1 | 0.609 |
| BMI (kg/m$^2$) | 26.6 [23.4–30.0] | 26.6 [23.2–30.7] | 0.817 | | 28.8 [25.0–32.6] | 28.9 [25.5–32.9] | 0.170 |
| **Race/Ethnicity (%)** | | | **0.026** | | | | **0.031** |
| White | 66.0 | 62.4 | | | 58.8 | 54.4 | |
| Black | 21.5 | 22.9 | | | 23.8 | 28.4 | |
| Hispanic | 8.0 | 10.1 | | | 10.8 | 11.1 | |
| Asian | 3.3 | 3.5 | | | 4.9 | 5.6 | |
| **Primary Payer (%)** | | | 0.613 | | | | 0.187 |
| Private | 51.1 | 50.5 | | | 44.7 | 48.8 | |
| Public | 48.1 | 48.4 | | | 54.7 | 50.3 | |
| **Cardiac Diagnosis (%)** | | | | | | | |
| Dilated Cardiomyopathy | 54.9 | 55.0 | 0.943 | | 45.4 | 49.5 | 0.056 |
| Restrictive Cardiomyopathy | 5.0 | 5.4 | 0.523 | | 2.3 | 2.9 | 0.370 |
| Ischemic Cardiomyopathy | 24.5 | 22.2 | **0.046** | | 43.5 | 38.8 | **0.028** |
| Congenital cardiomyopathy | 4.4 | 5.5 | 0.055 | | 0.9 | 1.6 | 0.189 |
| Hypertrophic cardiomyopathy | 4.1 | 3.3 | 0.123 | | 0.9 | 1.7 | 0.089 |
| Valvular | 1.2 | 1.4 | 0.473 | | 0.9 | 0.2 | **0.032** |
| **Cardiac Support at Time of listing (%)** | | | | | | | |
| Ventilator | 2.0 | 2.9 | **0.029** | | 1.8 | 2.2 | 0.473 |
| Inotropes | 36.3 | 36.6 | 0.791 | | 39.3 | 37.5 | 0.384 |
| LVAD | 26.3 | 20.8 | **<0.001** | | 31.8 | 24.3 | **<0.001** |
| RVAD +/- LVAD or MCS unspecified | 1.1 | 2.2 | **0.001** | | 1.2 | 1.4 | 0.735 |
| TAH | 0.3 | 0.4 | 0.644 | | 0.4 | 0.5 | 0.757 |
| ECMO | 2.7 | 4.7 | **<0.001** | | 1.2 | 3.0 | **0.005** |
| IABP | 5.6 | 16.9 | **<0.001** | | 6.2 | 16.8 | **<0.001** |
| **Comorbidities (%)** | | | | | | | |
| Tobacco user | 42.9 | 38.6 | **0.001** | | 48.3 | 46.5 | 0.384 |
| Malignancy | 8.3 | 9.1 | 0.335 | | 9.6 | 10.6 | 0.417 |
| Prior CVA | 6.2 | 6.7 | 0.503 | | 7.0 | 7.2 | 0.875 |
| ESRD | 2.9 | 3.4 | 0.335 | | 5.8 | 5.5 | 0.781 |
| AICD | 72.3 | 66.0 | **<0.001** | | 78.0 | 75.3 | 0.136 |
| Prior Cardiac Surgery | 37.2 | 35.1 | 0.098 | | 40.8 | 40.2 | 0.777 |
| **Outcomes (%)** | | | | | | | |
| Waitlist Time (IQR) | 47.0 [17.0–118.0] | 21.0 [7.0–70.0] | **<0.001** | | 47.0 [18.0–126.0] | 23.0 [8.0–73.5] | **<0.001** |
| **Transplant Outcomes (%)** | | | | | | | |
| Ischemic Time (IQR) | 3.0 [2.3–3.7] | 3.4 [2.8–4.0] | **<0.001** | | 3.1 [2.3–3.8] | 3.5 [2.8–4.0] | **<0.001** |
| Distance Traveled (IQR) | 82.0 [13.0–255.5] | 222.0 [80.3–398.8] | **<0.001** | | 85.0 [15.5–260.5] | 238.0 [83.5–404.5] | **<0.001** |

a higher mortality risk after transplantation even after controlling for demographic and comorbidity differences. Importantly, the 2018 allocation system change led to more equity in access to transplantation among patients with T2D. Despite improved access to

transplantation in the new allocation system though, patients with T2D continued to have differentially worse outcomes. The prevalence of T2D continues to increase in the US [6]. Between 1999 and 2016, the total percentage of adults living with T2D increased from 9.5% to 12.0% [7]. Our study demonstrates similar findings, where a growing proportion of patients listed for heart transplant had T2D. Demographically, T2D is a disease that afflicts all groups, but remains most prevalent in those who are male, Black or Hispanic, older than 65 years, and lower income [6,8], which is consistent with our findings.

Our study demonstrated increased mortality among transplanted patients with T2D, as well as a decreased likelihood of transplant. Previous literature on the association between T2D and post-transplant outcomes has been conflicting. Some studies demonstrate that patients with a preoperative diagnosis of T2D did not have worse mortality [5,9,10], while other studies indicated that pre-transplant T2D increased the likelihood of post-transplant mortality [2–4]. Interestingly, previous studies have demonstrated differences in transplant outcomes based not only on the level of diabetes control, but based on the presence of diabetes related complications. Specifically, Russo et al. found that with increasing levels of severe comorbidities, post-operative outcomes were worse for transplant patients with diabetes [11]. While previous studies have suggested the presence of additional diabetes complications as being the driver of worsened mortality, our study showed otherwise. Our study demonstrated that even when controlling for these comorbidities and more, outcomes of patients with T2D were significantly worse, albeit without controlling for glycemic control which was unavailable to us in this dataset. However, our study is the largest and most contemporary analysis of T2D in heart transplantation recipients. The data from this study could potentially be used to help guide clinician decisions and assist with risk stratification of patients listed for and receiving a heart transplant. Physicians should be aware of the increased risks of poor outcomes that diabetic patients face to provide more effective care. This increased awareness could improve outcomes and optimize resource allocation in this patient population. To better investigate the causes of worse outcomes, future research should examine the interplay of glycemic control and appropriate medical therapy with post-transplantation outcomes in patients with T2D. Additionally, further multi-institutional series could be able to better provide highly granular details regarding post-operative renal function or failure, vascular complications, infections, or acute or chronic rejection not available in the UNOS registry would provide meaningful insight for the transplantation community.

Following the allocation system changes in 2018, our study demonstrates that access to transplantation improved similarly for recipients with and without T2D. In 2018, the OPTN switched to a new allocation system in part to improve equity in organ allocation and enable broader distribution of donor hearts [12]. We found that both recipients with and without T2D were more likely to receive transplantation in the new allocation system. Interestingly, we found that when isolating only the new allocation system, there was no longer a significant difference in transplant likelihood between patients with and without T2D. This could be in part due to increased access of suitable donor organs at the cost of distance traveled and ischemic time. However, patients with T2D had a higher likelihood of mortality than patients without T2D and also saw an increase in mortality risk in the new system. Despite improved access to transplant, the continued increased mortality among recipients with T2D suggests that these outcomes are related to other patient level factors.

While T2D is rising in prevalence overall, there are worrisome signals that T2D is disproportionately affecting populations with socioeconomic disadvantage. In our study, we found that individuals listed for transplant with T2D were more likely to be non-White and have public insurance, an indicator of lower income given the financial requirements to obtain Medicaid in the United States. As a result, our findings suggest persistent concerns for the

sociodemographic equity of pre- and post-heart transplantation care and access. Given the socioeconomic associations of T2D and its control, the underserved populations which are disproportionately impacted by the T2D epidemic and which, independent of T2D, experience worse post-transplant survival [13–15], may require special consideration in the management of end stage heart failure. However, we believe our findings of worse post-transplantation outcomes should not preclude diabetic patients from heart transplant, given the benefits derived from treatment. Rather, it should stimulate vigorous efforts to identify modifiable factors for patients with T2D to improve transplant outcomes.

## Limitations

This study has multiple limitations, including its retrospective nature. This prevents conclusions of causation from being drawn. Secondly, there is no longitudinal data on more granular follow-up events that could be associated with T2D. As a result, post-transplantation analysis is limited to mortality. Importantly, there is no data on medication regimen and adherence for both insulin and oral medication for diabetes. Similarly, our data could not account for control of blood glucose levels, HbA1c, total duration of patient diabetes, and the development of diabetes after initial listing of patients. In addition, our data includes only patients who were listed for transplantation, and as such, would not include patients with more severe diabetes excluded from listing. Regarding the allocation system analysis, the new OPTN system has only been in effect for 2 years and therefore practices are continuing to evolve and may not be fully captured in this data set. Particularly, the effects of COVID-19 may not be fully captured or understood.

## Conclusions

Over the last ten years, the proportion of heart transplant recipients with T2D has increased. These patients are more likely to be from historically underserved populations and suffer from a greater comorbidity burden. Importantly, patients with T2D they are less likely to be transplanted post-transplant mortality risk, even after adjustment for confounders. However, following the OPTN allocation system change in 2018, access to hearts seems to have improved for patients with T2D. It is important for future research to explore further granularity in outcomes in this patient population and to investigate the effects of glycemic control on outcomes.

## Supporting information

**S1 File.**
(DOCX)

## Author Contributions

**Conceptualization:** Fouad Chouairi, Clancy W. Mullan, Ahmed Ahmed, Jasjit Bhinder, Avirup Guha, P. Elliott Miller, Ania M. Jastreboff, Maya Rose Chiravuri, Arnar Geirsson, Nihar R. Desai, Sounok Sen, Tariq Ahmad, Muhammad Anwer.

**Data curation:** Fouad Chouairi, Clancy W. Mullan, Ahmed Ahmed, Jasjit Bhinder, Avirup Guha, P. Elliott Miller, Michael Fuery, Sounok Sen, Tariq Ahmad.

**Formal analysis:** Fouad Chouairi.

**Investigation:** Fouad Chouairi, Tariq Ahmad.

**Methodology:** Fouad Chouairi, Clancy W. Mullan, P. Elliott Miller, Tariq Ahmad, Muhammad Anwer.

**Software:** Fouad Chouairi.

**Supervision:** Sounok Sen, Tariq Ahmad, Muhammad Anwer.

**Validation:** Fouad Chouairi, Ania M. Jastreboff.

**Visualization:** Fouad Chouairi.

**Writing – original draft:** Fouad Chouairi.

**Writing – review & editing:** Fouad Chouairi, Clancy W. Mullan, Ahmed Ahmed, Jasjit Bhinder, Avirup Guha, P. Elliott Miller, Ania M. Jastreboff, Michael Fuery, Arnar Geirsson, Nihar R. Desai, Christopher Maulion, Sounok Sen, Tariq Ahmad, Muhammad Anwer.

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
