## [Decision Letter · Decision Letter 0]

25 Feb 2022

PONE-D-22-01999Clinical Implications of Type 2 Diabetes on Outcomes after Cardiac TransplantationPLOS ONE

Dear Dr. Chouairi,

Thank you for submitting your manuscript to PLOS ONE. After careful consideration, we feel that it has merit but does not fully meet PLOS ONE’s publication criteria as it currently stands. Therefore, we invite you to submit a revised version of the manuscript that addresses the points raised during the review process.

ACADEMIC EDITOR: This paper addresses a topic of interest in the heart transplant community.

Three expert reviewers have expressed interest, but also highlighted several major concerns that will need to be addressed thoroughly in revisions that will undergo vigorous peer review again.

There seem to be missing elements in the data collection and analyses (see details below, but for example WL mortality of DM vs non-DM patients, analysis on NODAT, donor age, cause of death etc), a more thorough discussion about diabetic patients' mortality after heart Tx (not due to DM per se but due to complications of DM), information on how diabetes was treated needs to be included, and very importantly how this paper would change clinical practise in the view of the authors. Would also agree to embark on professional English editing of the MS.

We look forward to receiving your revised manuscript.

Kind regards,

Frank JMF Dor, M.D., Ph.D., FEBS, FRCS

Academic Editor

PLOS ONE

Journal Requirements:

2.Please provide additional details regarding participant consent. In the ethics statement in the Methods and online submission information, please ensure that you have specified (1) whether consent was informed and (2) what type you obtained (for instance, written or verbal, and if verbal, how it was documented and witnessed). If your study included minors, state whether you obtained consent from parents or guardians. If the need for consent was waived by the ethics committee, please include this information.

"I have read the journal's policy and the authors of this manuscript have the following competing interests: Dr. Desai works under contract with the Centers for Medicare and Medicaid Services to develop and maintain performance measures used for public reporting and pay for performance programs. He reports research grants and consulting for Amgen, Astra Zeneca, Boehringer Ingelheim, Cytokinetics, Medicines Company, Relypsa, Novartis, and SCPharmaceuticals. Dr. Jastreboff serves as a consultant for Novo Nordisk, Eli Lilly, Boehringer Ingelheim and receives research support from the American Diabetes Association, Eli Lilly, Novo Nordisk, and The NIH/ NIDDK. Dr. Ahmad is a consultant for Amgen, Cytokinetics, Relypsa, and Novartis. The other authors have no relationships." 

Reviewers' comments:

Reviewer's Responses to Questions

**Comments to the Author**

1. Is the manuscript technically sound, and do the data support the conclusions?

Reviewer #1: Partly

Reviewer #2: Yes

Reviewer #3: Yes

2. Has the statistical analysis been performed appropriately and rigorously? 

Reviewer #1: Yes

Reviewer #2: Yes

Reviewer #3: Yes

3. Have the authors made all data underlying the findings in their manuscript fully available?

Reviewer #1: Yes

Reviewer #2: Yes

Reviewer #3: Yes

4. Is the manuscript presented in an intelligible fashion and written in standard English?

Reviewer #1: No

Reviewer #2: Yes

Reviewer #3: Yes

5. Review Comments to the Author

Reviewer #1: The authors highlight an important topic in patients undergoing heart transplantation: diabetes mellitus. They have taken data from the UNOS database with 9086 patients with T2D and 23676 without T2D listed for transplantation and state the following:

- the proportion with heart transplantation recipients with T2D has increased

- these patients are more likely to be from underserved populations

- these patients have a lower likelyhood of transplantation and a higher likelyhood of post-transplantation mortality

- the change in allocation system increased their chances for transplantation

I have concerns regarding this paper:

Major:

1. The paper could do with editing by a English native speaker.

2. What is the difference in waiting list mortality between patients with and without T2D (conditional survival until transplant)? This is stated in the method section but I see now raw data on this.

3. What about the fact that some patients experience post-transplant DM early or late after transplantation (Sjarif et al, AJT 2014) and how does this affect your analysis? This is something that is not taken into account in the current analysis and should be corrected since it will affect outcomes.

4. There is no data given about how T2D was treated (pre- and post-transplant). This is of great importance since SGLT2 inhibitors and GLP-1 analogues have shown to have positive cardiometabolic effects.

5. The authors state that renal function is not available in the UNOS database. However, this is available. They should correct this.

6. The authors state that patients are more likely to die after transplantation. However, this should be rephrased. Chances for al of us dying are 100%. They mean that patients are more likely to die sooner. This is reflected in the survival curves which early on diverge but the seem to have a similar slope. The authors should reflect upon this in their paper.

Minor:

1. Page numbers are missing and ISHLT is not written out fully.

Reviewer #2: Nice paper investigating the prognosis of diabetic patients after heart transplantation and on the waiting list on a large dataset.

I have the following observations:

- do the Authors have any data about potential causes of death after heart transplantation? Generally, diabetic people have an higher rate of infection: is this data available?

- donor age is one of the most important prognostic factor after heart transplantation, but it has not been analyzed. This is an important study limitation and this data must be analyzed . Did the diabetic patients receive an organ from an older donor?

- do we have any data about the urgency status of the patients in the study ?

In the literature, it has been shown that diabetic patients have an increased risk of death after heart transplantation , but that this is not related to diabetes per se, but rather to the presence of diabetic complications (chronic kidney disease, peripheral vasculopathy, etc.). This aspect is dissected in the discussion by the Authors, but only marginally.

What are the clinical implications of the study? Should we reassess the listing for heart transplantation for selected diabetic patients?

Reviewer #3: It should be shortened. Unfortunately, there were a lot of limitations, i.e. no medications. Please, add in the results and the discussion a bit more information about the primary payer - it was mentioned but looked like unnecessary information.

Did you have information about the donors' age, sex and causes of death? if yes, please, add it.

6. PLOS authors have the option to publish the peer review history of their article (what does this mean?). If published, this will include your full peer review and any attached files.

Reviewer #1: No

Reviewer #2: **Yes: **Marco Masetti

Reviewer #3: **Yes: **Maria Simonenko

---

## [Author Response · Author response to Decision Letter 0]

13 Apr 2022

Response to Reviewers

Reviewer #1

I have concerns regarding this paper:

Major:

1. The paper could do with editing by a English native speaker.

Author Response: Thank you for this comment. The native English speaking co-authors re-read and edited the entire paper to ensure grammatical accuracy and flow. Unnecessary verbiage was removed and the flow of the overall paper was improved.

2. What is the difference in waiting list mortality between patients with and without T2D (conditional survival until transplant)? This is stated in the method section but I see now raw data on this.

Author Response: Thank you for the important comment. We agree that it would be beneficial to include raw data on waitlist mortality. As a result the following was added to the results section: “There was no significant difference in waitlist mortality when the raw, unadjusted data was analyzed (No T2D: 4.8%, T2D 5.1%, p=0.300). After adjustment there was no significant difference in waitlist death; however, patients with T2D had a significantly higher likelihood of post-transplant death (HR: 1.30 (CI: 1.21-1.40), p<0.001).”

3. What about the fact that some patients experience post-transplant DM early or late after transplantation (Sjarif et al, AJT 2014) and how does this affect your analysis? This is something that is not taken into account in the current analysis and should be corrected since it will affect outcomes.

Author Response: Thank you for this important comment. Unfortunately this is a weakness of the UNOS dataset as it does not contain data on the development of new diabetes after transplantation. As a result the following was added to the limitations section: “Similarly, our data could not account for control of blood glucose levels, HbA1c, total duration of patient diabetes, and the development of diabetes after initial listing of patients.”

4. There is no data given about how T2D was treated (pre- and post-transplant). This is of great importance since SGLT2 inhibitors and GLP-1 analogues have shown to have positive cardiometabolic effects.

Author Response: Thank you for the important comment. A limitation of the UNOS database is the fact that medications are no available as a datapoint in the UNOS database. This is addressed in the limitations as follows: “Similarly, our data could not account for control of blood glucose levels, HbA1c, total duration of patient diabetes, and the development of diabetes after initial listing of patients.”

5. The authors state that renal function is not available in the UNOS database. However, this is available. They should correct this.

Author Response: Thank you for this important comment. The reviewer is correct that the presence of pre-transplant end stage renal disease (ESRD) is present in the UNOS database as is the need for dialysis prior to transplant and post-operatively. We changed the text to better reflect this. Additionally in our modeling and our table 1 we included ESRD as a variable.

6. The authors state that patients are more likely to die after transplantation. However, this should be rephrased. Chances for all of us dying are 100%. They mean that patients are more likely to die sooner. This is reflected in the survival curves which early on diverge but the seem to have a similar slope. The authors should reflect upon this in their paper.

Author Response: Thank you for this comment. As requested by the reviewer, we clarified our writing and changed the phrase “more likely to die” to “had a higher mortality risk” in multiple instances throughout the discussion. One example being, “This study examined the impact of T2D among contemporary patients listed for heart transplantation and we found that patients with T2D were less likely to receive transplant and had a higher mortality risk after transplantation even after controlling for demographic and comorbidity differences.”

We deemed that this was more appropriate wording based on the following study. https://www.karger.com/Article/Fulltext/328916

Minor:

1. Page numbers are missing and ISHLT is not written out fully.

Author Response: Thank you for these comments. As a requested, page numbers were added and ISHLT was written out fully in the text.

Reviewer #2: 

Nice paper investigating the prognosis of diabetic patients after heart transplantation and on the waiting list on a large dataset.

I have the following observations:

1. Do the Authors have any data about potential causes of death after heart transplantation? Generally, diabetic people have an higher rate of infection: is this data available?

Author Response: Thank you for this important question. We agree that this could be important data to present in our study. As a result, we tabulated all causes of death with an incidence greater than 1.0% of total deaths and added them to S3 Table in the supplementary data. Patients with diabetes had a higher incidence of infection as a cause of death as suggested by the reviewer. 

2. Donor age is one of the most important prognostic factors after heart transplantation, but it has not been analyzed. This is an important study limitation and this data must be analyzed. Did the diabetic patients receive an organ from an older donor?

Author Response: Thank you for the comment. We agree that donor age is an important prognostic factor. As a result we added it to S1 Table in the supplementary data. Additionally the following was added to the results section of the paper: “Additionally, patient with T2D received hearts from significantly older patients (No T2D: 30 years vs. T2D: 31 years, p<0.001).”

3. Do we have any data about the urgency status of the patients in the study?

Author Response: Thank you for the important comment. We agree that is important to list data on the urgency status of patients in the study both at listing and at transplant. As requested, urgency/listing status was included as S1 Fig in the supplementary data.

4. In the literature, it has been shown that diabetic patients have an increased risk of death after heart transplantation, but that this is not related to diabetes per se, but rather to the presence of diabetic complications (chronic kidney disease, peripheral vasculopathy, etc.). This aspect is dissected in the discussion by the Authors, but only marginally.

Author Response: Thank you for this important comment. We agree that previous literature has demonstrated diabetes complications as a potential driver for worse transplantation outcomes. As a result, we edited our paper to read as follows in the discussion, “Interestingly, previous studies have demonstrated differences in transplant outcomes based not only on the level of diabetes control but based on the presence of diabetes related complications. Specifically, Russo et al. found that with increasing levels of severe comorbidities, post-operative outcomes were worse for transplant patients with diabetes.[11] While previous studies have suggested the presence of additional diabetes complications as being the driver of worsened mortality, our study showed otherwise. Our study demonstrated that even when controlling for these comorbidities and more, outcomes of patients with T2D were significantly worse, albeit without controlling for glycemic control which was unavailable to us in this dataset.”

5. What are the clinical implications of the study? Should we reassess the listing for heart transplantation for selected diabetic patients?

Author Response: As requested we further elucidated the value and clinical implications in the discussion in the following: “The data from this study could potentially be used to help guide clinician decisions and assist with risk stratification of patients listed for and receiving a heart transplant. Physicians should be aware of the increased risks of poor outcomes that diabetic patients face to provide more effective care. This increased awareness could improve outcomes and optimize resource allocation in this patient population.”

Reviewer #3: 

1. It should be shortened.

Author Response: Thank you for the comment. As requested by the reviewer, we shortened the article by eliminated unnecessary verbiage in the discussion, results, and methods section in addition to improving the grammar in this paper as requested by reviewer 1.

2. Unfortunately, there were a lot of limitations, i.e. no medications. Please, add in the results and the discussion a bit more information about the primary payer - it was mentioned but looked like unnecessary information.

Author Response: Thank you for this comment. We added the following to the methods section of the paper: “Patients with T2D were more likely to be on public insurance and less likely to have private insurance (p<0.001)”. In terms of an explanation the following was added to the discussion section: “In our study, we found that individuals listed for transplant with T2D were more likely to be non-White and have public insurance, an indicator of lower income given the financial requirements to obtain Medicaid in the United States. As a result, our findings suggest persistent concerns for the sociodemographic equity of pre- and post-heart transplantation care and access.”

3. Did you have information about the donors' age, sex and causes of death? if yes, please, add it.

Author Response: Thank you for this important comment. Donor age, sex, and demographics were added in S1 Table in the supplementary material. Additionally, donor cause of death was added to the S1 Table.

---

## [Decision Letter · Decision Letter 1]

25 May 2022

PONE-D-22-01999R1Clinical Implications of Type 2 Diabetes on Outcomes after Cardiac TransplantationPLOS ONE

Dear Dr. Chouairi,

Thank you for submitting your manuscript to PLOS ONE. After careful consideration, we feel that it has merit but does not fully meet PLOS ONE’s publication criteria as it currently stands. Therefore, we invite you to submit a revised version of the manuscript that addresses the points raised during the review process.

ACADEMIC EDITOR:The authors unfortunately haven't fully addressed the issues raised by the reviewers in the MS itself, and have decided to provide the requested information in the supplements.

Donor age is a crucial prognostic factor in hear transplantation, and has not been analysed in multivariate analysis.

These crucial changes have to be made in the MS in order for the MS to be reconsidered.

We look forward to receiving your revised manuscript.

Kind regards,

Frank JMF Dor, M.D., Ph.D., FEBS, FRCS

Academic Editor

PLOS ONE

Reviewers' comments:

Reviewer's Responses to Questions

**Comments to the Author**

1. If the authors have adequately addressed your comments raised in a previous round of review and you feel that this manuscript is now acceptable for publication, you may indicate that here to bypass the “Comments to the Author” section, enter your conflict of interest statement in the “Confidential to Editor” section, and submit your "Accept" recommendation.

Reviewer #2: All comments have been addressed

Reviewer #3: All comments have been addressed

2. Is the manuscript technically sound, and do the data support the conclusions?

Reviewer #2: Yes

Reviewer #3: Yes

3. Has the statistical analysis been performed appropriately and rigorously? 

Reviewer #2: Yes

Reviewer #3: Yes

4. Have the authors made all data underlying the findings in their manuscript fully available?

Reviewer #2: (No Response)

Reviewer #3: Yes

5. Is the manuscript presented in an intelligible fashion and written in standard English?

Reviewer #2: (No Response)

Reviewer #3: Yes

6. Review Comments to the Author

Reviewer #2: Thank you for having partly answered to my questions. Howvever, I notice that the answers have been provided in the supplementary material section.

Reviewer #3: (No Response)

7. PLOS authors have the option to publish the peer review history of their article (what does this mean?). If published, this will include your full peer review and any attached files.

Reviewer #2: No

Reviewer #3: **Yes: **Maria Simonenko

---

## [Author Response · Author response to Decision Letter 1]

20 Jun 2022

Response to Reviewers:

Editor:

1. The authors unfortunately haven't fully addressed the issues raised by the reviewers in the MS itself, and have decided to provide the requested information in the supplements. Donor age is a crucial prognostic factor in hear transplantation, and has not been analysed in multivariate analysis. These crucial changes have to be made in the MS in order for the MS to be reconsidered.

Author Response: Thank you for these important recommendations. We agree that it is important to highlight the issues brought up by reviewer 2 in the manuscript itself. As a result, the majority of the supplemental data has been added to the manuscript itself. Specifically, S1 Table, S2 Table, and S3 Table have been converted into Table 2, Table 4, and Table 5 respectively. 

Additionally, we agree that donor age is a very important prognostic factor that should be included in our analysis of post-transplant mortality. As a result, we added donor age to every multivariate regression examining post-transplant mortality. These changes can be found in Table 3 and in the 2nd and 3rd sub-sections of the results sections.

Reviewer 2:

1. Thank you for having partly answered to my questions. However, I notice that the answers have been provided in the supplementary material section.

Author Response: Thank you for these important recommendations. We agree that it is important to highlight the issues brought up by reviewer 2 in the manuscript itself. As a result, the majority of the supplemental data has been added to the manuscript itself. Specifically, S1 Table, S2 Table, and S3 Table have been converted into Table 2, Table 4, and Table 5 respectively.

---

## [Decision Letter · Decision Letter 2]

3 Aug 2022

Clinical Implications of Type 2 Diabetes on Outcomes after Cardiac Transplantation

PONE-D-22-01999R2

Dear Dr. Chouairi,

We’re pleased to inform you that your manuscript has been judged scientifically suitable for publication and will be formally accepted for publication once it meets all outstanding technical requirements.

Kind regards,

Frank JMF Dor, M.D., Ph.D., FEBS, FRCS

Academic Editor

PLOS ONE

Additional Editor Comments (optional):

All comments addressed

Reviewers' comments:

Reviewer's Responses to Questions

**Comments to the Author**

1. If the authors have adequately addressed your comments raised in a previous round of review and you feel that this manuscript is now acceptable for publication, you may indicate that here to bypass the “Comments to the Author” section, enter your conflict of interest statement in the “Confidential to Editor” section, and submit your "Accept" recommendation.

Reviewer #3: All comments have been addressed

2. Is the manuscript technically sound, and do the data support the conclusions?

Reviewer #3: Yes

3. Has the statistical analysis been performed appropriately and rigorously? 

Reviewer #3: Yes

4. Have the authors made all data underlying the findings in their manuscript fully available?

Reviewer #3: Yes

5. Is the manuscript presented in an intelligible fashion and written in standard English?

Reviewer #3: Yes

6. Review Comments to the Author

Reviewer #3: (No Response)

7. PLOS authors have the option to publish the peer review history of their article (what does this mean?). If published, this will include your full peer review and any attached files.

Reviewer #3: **Yes: **Maria Simonenko

---

## [Editor Report · Acceptance letter]

11 Aug 2022

PONE-D-22-01999R2 

Clinical Implications of Type 2 Diabetes on Outcomes after Cardiac Transplantation 

Dear Dr. Chouairi:

I'm pleased to inform you that your manuscript has been deemed suitable for publication in PLOS ONE. Congratulations! Your manuscript is now with our production department. 

Kind regards, 

on behalf of

Dr. Frank JMF Dor 

Academic Editor

PLOS ONE